# Comparative Genomics and Pan-Genome Driven Prediction of a Reduced Genome of *Akkermansia muciniphila*

**DOI:** 10.3390/microorganisms10071350

**Published:** 2022-07-04

**Authors:** Sayyad Ali Raza Bukhari, Muhammad Irfan, Irfan Ahmad, Lijing Chen

**Affiliations:** 1Department of Biotechnology, University of Sargodha, Sargodha 40100, Pakistan; aliraza3119@gmail.com; 2Department of Clinical Laboratory Sciences, College of Applied Medical Sciences, King Khalid University, Abha 61421, Saudi Arabia; irfancsmmu@gmail.com; 3Key Laboratory of Agriculture Biotechnology, College of Bioscience and Biotechnology, Shenyang Agricultural University, Shenyang 110866, China

**Keywords:** comparative genomics, pan genomics, horizontal gene transfer, probiotics, evolution, population structure, next-generation probiotics, minimal genome

## Abstract

*Akkermanisia muciniphila* imparts important health benefits and is considered a next-generation probiotic. It is imperative to understand the genomic diversity and metabolic potential of the species for safer applications as probiotics. As it resides with both health-promoting and pathogenic bacteria, understanding the evolutionary patterns are crucial, but this area remains largely unexplored. Moreover, pan-genome has previously been established based on only a limited number of strains and without careful strain selection. The pan-genomics have become very important for understanding species diversity and evolution. In the current study, a systematic approach was used to find a refined pan-genome profile of *A. muciniphila* by excluding too-diverse strains based on average nucleotide identity-based species demarcation. The strains were divided into four phylogroups using a variety of clustering techniques. Horizontal gene transfer and recombination patterns were also elucidated. Evolutionary patterns revealed that different phylogroups were expanding differently. Furthermore, a comparative evaluation of the metabolic potential of the pan-genome and its subsections was performed. Lastly, the study combines functional annotation, persistent genome, and essential genes to devise an approach to determine a minimal genome that can systematically remove unwanted genes, including virulent factors. The selection of one strain to be used as a chassis for the prediction of a reduced genome was very carefully performed by analyzing several genomic parameters, including the number of unique genes and the resistance and pathogenic potential of the strains. The strategy could be applied to other microbes, including human-associated microbiota, towards a common goal of predicting a minimal or a reduced genome.

## 1. Introduction

*Akkermansia muciniphila* is an anaerobic, Gram-negative bacterium having great potential as a probiotic representing 3% of intestinal bacteria as found in adult feces [1]. It is specialized in utilizing gut mucins as nitrogen and carbon sources. Although the bacterium is a strict anaerobic microbe, recently its ability to colonize near epithelial cells—characterized by microaerophilic levels of oxygen—has been determined [2]. It cannot synthesize the most abundant amino acid in its structure—threonine—and obtains it externally by degrading mucin [3,4]. Mucin degradation drives the production of short-chain fatty acids (SCFAs)—such as propionate, acetate, and butyrate—close to the epithelial cells [5]. The spatial location of SCFAs’ production makes them readily available to the host [6]. The production of SCFAs induces a signaling response in the host via the Gpr43 receptor resulting in stimulation of immune response in monoassociated germ-free mice [6,7,8]. Furthermore, studies have suggested a positive association between *A. muciniphila* and intestinal health. Due to widespread health impacts, it is considered a biomarker for a healthy intestine [9,10]. Among the mucosa-associated microbes, *A. muciniphila* has been found to be comparatively higher in healthy controls as compared to patients with irritable bowel disease [11]. The comparative scarcity of *A. muciniphila* was found to be related to appendicitis [6], obesity [12], and autism [13].

The potential health benefits of probiotics have attracted much interest in engineering more useful and robust microbes that can produce a variety of beneficial metabolites and fight against pathogens [14,15]. The use of probiotics is an innovative and safer approach to treating infectious diseases compared to antibiotics due to the risk of multi-drug resistance development in pathogenic bacteria. Probiotics have different mechanisms to inhibit pathogens such as secretion of antibacterial compounds, competition of space and nutrition, activation of the immune response, and inhibition of virulent factors in pathogens [16,17,18]. Many pathogenic bacteria develop biofilms that impart resistance against antibiotics, administration of engineered probiotics has demonstrated a positive response to inhibit pathogen growth in such circumstances [19,20]. The studies have suggested the effectiveness of engineered probiotics against *Salmonella* and *Vibrio cholera* [21,22,23].

In a natural setting, microbes are subject to an adaptive evolution that is driven by various forces such as horizontal gene transfer (HGT), gene duplication, and interaction among communities [24,25]. If the newly acquired genes are advantageous for a favorable trait, they undergo directional selection and sweep through the population over time [26]. The horizontally acquired genes are often organized into genomic islands. *A. muciniphila* resides with a dense community of microbes driving several evolutionary forces to act upon and result in increasing diversity. The gene gain/loss and genomic diversification in strains have given rise to the concept of pan-genomics.

The pan-genome refers to the complete set of genes present in a collection of organisms [27]. Pan-genome is divided into the core genome, accessory genome, and unique genome. The core genome reflects the pool of genes shared by at least 99% of the strains. Genes that are shared by more than 95% but less than 99% of the strains are called the soft-core genome. In order to incorporate the effect of missing genomic information due to partial genome assembly, the core and soft-core genomes have been combined and collectively referred to as persistent/conservative genome: referred to genes shared by more than 95% of the strains/genomes [28]. Persistent genes are the ones that have stably established in the whole population and plausibly evolved to impart essential or important functions in the population such as basic biology and phenotype of the species. Accessory genes, on the other hand, are the collection of genes that are shared among two or more genomes but less than the persistent genome threshold [29]. These genes potentially impart unique characteristics to a strain and/or provide a niche-specific advantage to the host strains. These genes may be acquired by horizontal gene transfer (HGT) and are maintained by a subset of all the strains of a species. The accessory genome is often split into shell genome (genes shared by more than 15% of the strains but less than 95% of the strains) and cloud genome (genes shared by less than 15% of the strains but more than one strain). Unique genes are those that are found in only a single genome/strain. These may be very recently diverged or obtained via HGT. The survival of these genes in the population is unpredictable. In this analysis, gene sets were structured as persistent genes, shell genes, cloud genes, and unique genes.

Prediction of a stable genome with an enhanced expression of desired pathways is critical in synthetic biology for the synthesis of robust next-generation probiotics. Intense efforts are being made, currently, to synthesize bacteria with reduced stable genomes and desired properties. Redundancy in natural organisms precludes a clear understanding of the functional properties of an organism. Furthermore, reduced genomes have less burden of extra genes and perform specific tasks more efficiently. The central objective of synthetic biology is to determine a “minimal gene set”; however, the “minimal gene set” is highly variable depending not only on the species but also on environmental factors [30]. For a stable synthetic genome, the concept of the minimal genome is elusive and needs the incorporation of additional genes that provide additional benefits in the context of varying environments. Several attempts have been made to delineate a minimal genome; however, considering varying environmental pressures, the efforts are unproductive for the long-term survival of the genome [31,32,33]. For the long-term stability under fluctuating conditions, the concept of a “persistent genome” has been introduced [33].

A second important factor in probiotic usage is the strain-specific probiotic potential. It has previously been reported that the probiotic potential of an organism is strain-specific and cannot be generalized to the whole species [34]. In this regard, a separate comprehensive analysis of probiotic potential is necessary before human consumption. More specifically, a strain-specific functional analysis focusing on antibiotic resistance, pathogenicity determinant genes, and antimicrobial peptides is important. It will be vital for the suitable selection of a chassis strain for the reduction of the genome. Therefore, all strains were used to determine their resistance and pathogenic potential for the selection of a chassis strain on which a strategy to predict a reduced genome was applied.

Pan-genomics has diverse applications in microbial genomics such as taxonomy, evolutionary dynamics, strain diversity, strain identification, and reverse vaccinology among many others. However, the number and choice of strains strongly impact the accuracy of the downstream analysis [35]. Previously, a very limited number of strains of *A. muciniphila* have been used for the pan-genome analysis, which is insufficient for a true representation of pan-genome. Furthermore, the evolutionary dynamics of *A. muciniphila* has not been elucidated. We performed a comprehensive analysis for the prediction of a reduced genome that can be used as a safe and robust synthetic probiotic. Firstly, strains were selected for pan-genome analysis using average nucleotide identity and gene presence-absence pattern. Secondly, functional annotation of all the strains was performed to select a suitable strain that can act as a chassis for the genome reduction strategy. Lastly, we combined functional annotation, pan-genomics, and essentiality analysis to suggest a reduced genome as a safe and robust synthetic probiotic. The strategy is also helpful for better understanding the molecular biology of probiotics. It can be applied to other microbes as well that provide health benefits to humans and animals.

## 2. Materials and Methods

### 2.1. Data Collection

All the available genomes of *A. muciniphila* were retrieved from the NCBI RefSeq database using a custom bash script on 24 March 2021. The genomes were renamed from assembly names to strain names for easy differentiation among different genomes.

### 2.2. Quality Assessment, Gene Annotation and Average Nucleotide Identity

The gene prediction was performed using Prodigal 2.6 [36] as implemented in Prokka v1.12 [37]. The latter was also used to implement pipelines for the prediction of tRNA and tmRNA, rRNA, and CRISPR sequences using Aragorn [38], Barrnap v0.8 (https://github.com/tseemann/barrnap, accessed on 18 June 2017), and MinCED v0.4.0 (https://github.com/ctSkennerton/minced, accessed on 28 March 2019), respectively.

Average nucleotide identity (ANI) was performed to explore the taxonomic boundary of the genomes using pyani v0.2.10 (https://github.com/widdowquinn/pyani accessed on 4 June 2020) with BLASTP [39,40] as a similarity search program. The genomes were also subjected to quality control as implemented in Panaroo [41]. As part of quality control, Panaroo finds Mash distances between genomes using Mash [42]. It provides a genome-scale dissimilarity score that can be converted to similarity scores equivalent to average nucleotide identity.

### 2.3. Pan-Genome Estimation and Modeling

The pan-genome analysis was performed on both the total strains and shortlisted strains. Shortlisting was performed based on average nucleotide identity, Mash distances, and gene presence-absence. The pan-genome analysis was performed by Panaroo using the sensitive algorithm of the software. The pan-genome was plotted using roary_plots.py v0.1.0 (https://github.com/sanger-pathogens/Roary/blob/master/contrib/roary_plots/roary_plots.py accessed on 28 June 2020). The pan-genome was modeled using PanGP [43]. The core genome was aligned using MAFFT—a multiple sequence alignment tool [44]. GC content and gene length of the pan-genome were calculated using SeqKit [45].

The size and trajectory of the pan- and core genomes were estimated using models described previously [27,35]. The pan-genome profile was modeled using a power-law regression model based on Heap’s law [y = A_pan_x^Bpan^ + C_pan_] as described by Tettelin et al. [35] and implemented in PanGP [43]. Here, y represents the size of the pan-genome, x represents the genome number, and Apan, Bpan, and Cpan are fitting parameters. Among the fitting parameters, Bpan is equivalent to the parameter γ used by Tettelin et al. [35] in estimating the open or closed nature of a pan-genome. The core genome modeling was performed using exponential regression described by Tettelin et al. [35] and implemented in PanGP [43].

### 2.4. Phylogenetic Analysis and Population Structure

The core genome multiple sequence alignment was used for the inference of phylogenetics using FastTree 2.1 [46] and RAxML [47]. GTR + CAT algorithm was used for the construction of the phylogenetic tree. The tree was visualized using a Bioconductor package ggtree [48].

The core genome SNPs were detected using Snippy (https://github.com/tseemann/snippy accessed on 25 March 2021). The recombination sites were filtered using Gubbins [49]. The recombination filtered alignment was also used for the inference of phylogenetic trees using FastTree 2.1 [46]. GTR + CAT model was used for phylogenetic analysis.

The population structure was analyzed using PopPUNK [50]. The PopPUNK finds the core genome distances and accessory genome distances to cluster genomes into distinct groups.

The correlation of the accessory genome was carried out by converting the gene presence-absence profile to Jaccard distances using R [51]. The distance matrix was used for principal component analysis and visualized to assess the clustering of genomes.

### 2.5. Functional Annotation and Enrichment Analysis

The pan, persistent, shell, cloud, and unique genes were annotated using KAAS [52] and EggNOG mapper [53]. Antibiotic resistance genes and virulent factors were determined by ABRicate v0.8 (https://github.com/tseemann/abricate accessed on 16 July 2018). Antimicrobial peptides and secondary metabolite biosynthesis gene clusters were determined via antiSMASH [54] and BAGEL4 [55]. To explore the statistical significance of the functional annotation, clusterProfiler [56] was used. clusterProfiler requires an organism database of the species in question; however, the database of *A*. *muciniphila* was missing. It was developed using the Bioconductor package AnnotationForge [57]. Pan-genome annotation by EggNOG mapper was used for building the database.

To assess the antibiotic resistance genes and virulent factors in the strains, ABRicate was employed using VFDB and NCBI databases, respectively. A minimum coverage of 30% and a minimum sequence identity of 75% were used for the analysis. Pathogenicity islands were predicted in *A. muciniphila* ATCC BAA835 using IslandViewer 4 [58].

### 2.6. Evolutionary Analysis

To test whether different sections of the pan-genome are exhibiting different selection, the non-synonymous to synonymous substitution ratio was calculated using Ka/Ks Calculator. Unique genes were excluded in this test, as the analysis requires genes from at least two organisms.

Gubbins [49] was used to predict the high recombination rate segments. The recombination events were visualized on the Phandango web server (accessed on 20 October 2020) [59]. To determine HGT genes, HGTector [60] was used using BLASTP as a similarity search tool.

### 2.7. Essentiality Analysis

To determine the essentiality of the genes, all the genes in the pan-genome were compared to the database of essential genes (DEG) [61]. Similarly, all the genes of *A. muciniphila* ATCC BAA-835 were compared to DEG. The comparison was performed using BLASTP for protein-coding genes and BLASTN for non-protein-coding genes. Sequence identities of 30% and 70% were considered for BLASTP and BLASTN, respectively, with an e-value of 10^−3^. One best hit was considered for each blast event.

## 3. Results

### 3.1. General Characteristics of the Genomes

A total of 149 *A. muciniphila* genomes available in RefSeq (on 24 March 2021) were retrieved. The number of genes ranged from a minimum of 2140 in *A. muciniphila BSH 01* to a maximum of 2738 in *A. muciniphila BIOML A18* and a mean and median of 2353 and 2323, respectively. Concerning the number of contigs, 75% had less than or equal to 24 contigs. The highest contigs number observed was 237 in *A. muciniphila BIOML A17*.

In the 130-genome dataset, obtained after filtering out 19 genomes as discussed below, the average gene content in genomes was 2335 and the persistent gene set (core + softcore) was 1906. On average, 430 genes in addition to persistent genes are part of a single genome. All the genomic features are available in Appendix A.

### 3.2. Average Nucleotide Identity (ANI) and Mash Distances

Two related criteria were used to determine species boundaries—ANI and Mash distances. A standard cut-off of >95% ANI was considered a species boundary [62,63,64]. Among the 149 strains, 19 strains showed less than 95% ANI. Mash distances were also converted to similarity scores. The Mash distance scores showed the same trend. The same 19 strains showed less than 95% Mash similarity. It was also observed that the gene presence-absence of these 19 strains was too diverse from the remaining strains. This observation supported the idea of excluding these 19 strains. The approach used to exclude 19 strains has been demonstrated in Figure 1.

### 3.3. Pan-Genomics: Open Pan-Genome, Large Core and Unique Gene Sets

Pan-genomics of the whole dataset was performed, followed by plotting the pan genome matrix. The matrix disclosed the deviance in the presence-absence profile of the dissimilar 19 strains from the rest of the matrix. The matrix supported the ANI and Mash distances analysis that these 19 strains deviate enough from the other strains. Making it plausible to remove them from the downstream analysis in the quest for a robust pan-genome.

After the removal of the 19 strains, 130 strains remained. The dataset of 130 strains was again used for the construction of the pan-genome using the same parameters. The pan-genome of 130 strains lead to an increase of 351 core genes and a decrease of 139, 234, and 1852 in the soft-core, shell, and cloud genes, respectively. The total number of genes decreased from 9499 to 7625 in 130 strains as compared to 149 strains as demonstrated in Figure 2. Excluded 19 genomes were impacting the pan-genome by decreasing the core genome and incorporating a significant number of unique genes.

Heap’s law modeling of the gene presence-absence revealed that the *A. muciniphila* is an open genome with a γ value of 0.49. It suggests that the species is not saturated and the addition of new genomes into the dataset will result in an increase in the pan-genome (total number of genes in the species). For γ < 0, the pan-genome of the dataset is considered closed, and the total number of genes do not increase as more and more strains are added to the species, whereas for γ > 0, the pan-genome is open, and its size increases as more and more strains sequences are included in the species. Fitting parameters can be visualized in Figure 3A.

### 3.4. Population Structure

Multiple methods were applied to elucidate the population structure. Average nucleotide identity measurements were subjected to different machine learning models such as hierarchical clustering, partitioning around medoids, and K-means clustering to understand population structure. Total within the sum of square, gap distances, and average silhouette width were employed to understand strain clustering. It was noticed that most of the algorithms structured the strains into four clusters. Visual observation of the plots for these algorithms suggested three or four as potential clusters. Silhouette width reflected that three clusters solution leave one strain with negative silhouette width, indicating inappropriate cluster assignment of that strain. The average silhouette width was found to be 0.71. Four cluster solutions improved the average silhouette width to 0.73 and supported all the clusters without any negative value in the dataset. The visual representation of all the machine learning models is represented in Figure 4.

Furthermore, to resolve the ambiguity in the above methods, PopPUNK was employed to calculate the core distance (α) and accessory distances (π). These α and π distances were found to be scattered from 0–0.35 and 0–0.03, respectively. Although π were spread across the range, α was clustered around 0, 0.008, 0.012, 0.017, and 0.0225. An optimum number of clusters was found to be four. Trying to fit data into three clusters merged the central two clusters, but five clusters resulted in an overlap of two clusters. Furthermore, the principal component analysis (PCA) and tSNE (Appendix A) of the gene presence-absence profile produced four distinct clusters. It is important to note that gene presence-absence corresponds to the ANI and PopPUNK. PopPUNK output and PCA plot are also demonstrated in Figure 4.

### 3.5. Phylogroups Are Expanding at Varying Rates

In order to estimate the gene expansion within clades, Heap’s law (power-law) and exponential regression were applied to the pan- and core genomes of each phylogroup, respectively. The γ values were 0.31, 0.17, 0.2, and 0.9 for AmI, AmII, AmIII, and AmIV, respectively. It demonstrates that all the phylogroups had different trajectories for possible pan-genome expansion. AmIV showed higher trajectory than other phylogroups while AmII was most saturated among all as demonstrated by the flattened trajectory in Figure 3C. It was visible that the γ values of phylogroups were independent of the within-group ANI value—higher ANI did not result in higher diversity (Figure 3B).

### 3.6. Metabolic Potential of A. muciniphila

*A. muciniphila* has diverse metabolic potential. Many probiotics are characterized by their ability to ferment non-digestible fibers into short-chain fatty acids (SCFAs) such as n-butyrate, acetate, and propionate. SCFAs are important gut microbial products that can impact multiple host processes including energy consumption, microbe-host signaling, and decreased colonic pH. These processes, in turn, affect the gut microbial composition, gut motility, and epithelial cell proliferation [65]. SCFAs act as both a local nutrient source for colonocytes and a nutrient source for different microbes in the gastrointestinal tract [66]. An upshot of an eight to ten-fold increase in serotonin has been observed in an in-vitro colonic mucosal system by treatment of SCFAs [67]. Two different pathways have been observed in *A. muciniphila* for the synthesis of propionate—the malonate semialdehyde pathway and the propanoyl-CoA metabolism. Similarly, an extensive network of genes involved in methane metabolism has been identified using KEGG pathways.

The current study observed that the *A. muciniphila* genomes anabolize gut-derived lipopolysaccharides (LPS) also known as endotoxins. Two types of LPSs have been observed—*N*-acetylneuraminic acid and *N*-acetyl-beta-L-fucosamine. LPSs are immunostimulants and induce mild inflammation. In addition to LPSs, *A. muciniphila* is capable of producing a variety of glycans including O-antigen nucleotide sugar biosynthesis and other O-linked and N-linked glycans.

Genes for the biosynthetic pathways of thiamine, nicotinamide adenine dinucleotide (NAD+), pantothenate, biotin, and lipoic acid have been found to be part of the persistent genome. The pathways for tetrahydrofolate, Coenzyme A, riboflavin, cobalamin, and heame synthesis have one missing enzyme. All the pathways and the corresponding number of genes observed using KEGG are represented as a bar plot in Figure 5. The metabolic potential of all the subsections of the pan-genome is available in Appendix A.

The persistent, shell, cloud, and unique genomes were also annotated using clusters of orthologous (COG) database. The top five categories of significant similarity are “Function unknown”, “Replication/recombination and repair”, “Cell wall/membrane/envelop biogenesis”, “Amino acid transport and metabolism” and “Transcription”. The shell and cloud genomes have a higher proportion of genes in “Replication/recombination and repair”, “Cell wall/membrane/envelop biogenesis”, “Transcription” and “Defense mechanisms” as compared to the persistent genome. It depicts the role of the shell and cloud genomes in transcription and defense mechanisms. Some of the important categories where the persistent genome has a higher proportion of genes as compared to other sections of the pan-genome are “Amino acid transport and metabolism”, “translation/ribosomal structure and biogenesis”, “carbohydrate transport and metabolism”, and “energy production and conversion”. All the COG categories have been sorted according to the total number of gene counts and plotted in Figure 6A. Furthermore, core genes of the phylogroups were annotated by the EggNOG mapper and compared with each other and with the core genes of the complete data set (130 strains). It reflected that for each COG category, core genes of the phylogroups, as well as the complete data set, encode an almost equal number of genes Figure 6B.

### 3.7. Functional Enrichment of Genes

Functional enrichment of each section of the pan-genome (persistent, shell, cloud, and unique) was performed. It was observed that none of the sections entail enriched genes except the persistent genome. Most enriched genes belonged to ribosome synthesis and cytoplasm. The enrichment analysis has been demonstrated in Figure 6C.

### 3.8. Virulent Factors, Antibiotic Resistance Genes

All 130 strains were predicted for the presence of antibiotic resistance genes and virulent factors. It was observed that out of 130 strains, 23 contain at least one antibiotic resistance gene. *A. muciniphila GP38* contains the maximum number (four) of antibiotic-resistant genes. It contains two types of aminoglycoside (aph Id and aph Ib), tetracycline resistance, and sulfonamide resistance. The majority of the strains (15) are resistant to the lincosamide class of antibiotics followed by tetracycline (5) and macrolide (4). *A. muciniphila* BIOML-A13 contains a pathogenic gene secreted auto transport toxin with 99.64% identity over 35.21% coverage. Antibiotic resistance genes and virulent factors are shown in Figure 7. Genome lengths, GC contents, the presence of unique genes in each strain, and the presence of CRISPR sequences have also been shown around a phylogenetic tree in Figure 7. It is evident from Figure 7 that AmIII and AmIV have the least number of antibiotic resistance genes and pathogenicity factors. Similarly, most strains of AmIII and many members of AmIV and AmII have saturated genomes with fewer unique genes, reflecting the stability of the genomes. This is helpful in screening for the chassis strain for the genome reduction. We selected ATCC BAA-835 as a chassis for several reasons such as complete assembly, absence of antibiotic resistance, absence of virulent factor, no unique genes, and lastly, it is a representative genome.

### 3.9. Functionally Annotated Genes within Each Group Express Different GC Content

The median GC content and gene length were found to be higher in persistent genes as compared to the shell, cloud, and unique genes. The majority of the persistent genes had a GC content of around 58% and gene length of around 900 base pairs. The median GC content and gene length of all other sections were around 50% and 600 base pairs, respectively, as demonstrated in Figure 8. Among the persistent, shell, cloud, and unique gene sets, GC content and gene length were statistically significant except for the gene length between the shell and cloud genomes. All four groups were annotated by KEGG and EggNOG mapper. Among different groups, 54%, 13%, 10%, and 22% of the genes were successfully annotated by KAAS in persistent, shell, cloud, and unique genes, respectively. Within each group, GC content and gene length were significantly different between annotated and non-annotated gene sets. It was observed that the majority of the annotated genes tend to have higher GC content and/or gene length compared to non-annotated genes that were shorter in size or lower in GC percentage as demonstrated in a 2d density plot in Figure 8. In each of the four sections of the pan-genome, the median gene length and GC percentage were higher in annotated genes as compared to non-annotated genes as demonstrated in Figure 8. The results suggest a role of GC content and gene length in the evolution and functional repertoire of the organism. The genes that had high GC content and higher gene length were maintained in the species and were playing a functional role in the genome. The low GC content genes and small-sized genes were either easier to obtain through horizontal gene transfer and/or difficult to maintain in all strains of the species. These were also less likely to have a functional role in a genome.

### 3.10. Recombination Analysis

Very extensive recombination has been observed among the strains. Recombination is more prevalent within a phylogroup; however, the occurrence of recombination across phylogroups is evident. Recombination across the pan-genome has been visualized using Phandango [59]. AmIV is the largest phylogroup and exhibits the most extensive recombination pattern among all the phylogroups, as shown in Figure 9A. A network of recombination pattern was constructed using igraph package in R. The node indicates the strain name, and the edge represents recombination between the nodes. The nodes are colored by phylogroups. The network graph revealed that all the phylogroups share recombination; however, some strains of the phylogroup AmII form a distinct cluster. There is one exception that one strain of AmIV shares recombination events with a distinct cluster of AmII as indicated in the lower cluster in Figure 9B.

### 3.11. Horizontal Gene Transfer

Around 15% (1164) of the genes in the pan-genome are derived via HGT. Among the HGT genes, the percentages of the persistent, shell, cloud, and the unique genome are 11.4%, 5%, 15.8%, and 67.8%, respectively, while out of non-HGT genes, the percentages of the persistent, shell, cloud, and unique genomes are 27.6%, 8.7%, 30.3%, and 33.5%, respectively. Similarly, among the unique, cloud, shell, and persistent genomes, the percentages of HGT are 26.84%, 8.63%, 9.4%, and 6.98%, respectively Figure 10A. The majority of HGT genes were found in other gut microbes such as Clostridiales, Bacteroidales, Firmicutes, and Akkermansiaceae Figure 10B. However, the direction of gene transfer is unknown.

### 3.12. Genomic Islands, Prophages and Secondary Metabolite Gene Clusters

For secondary metabolite gene clusters, presence of prophages, and genomic islands, representative strain *A. muciniphila* ATCC BAA-835 was used. Five different genomic islands across *A. muciniphila* ATCC BAA-835 were predicted using IslandViewer 4 [58] as shown in Figure 10C. Gene clusters related to terpene and arylpolyene were predicted by antiSMASH, and sactipeptide was predicted by BAGEL4.

### 3.13. Non-Synonymous to Synonymous Mutation Calculation

The genes of the persistent, shell, and cloud genomes were used to find non-synonymous to synonymous mutation (Ka/Ks) calculations. Ka/Ks was observed to be lowest in the persistent genome and increased sequentially in the shell and cloud genome. High Ka/Ks values in shell and cloud genomes denote evolutionary pressure on the gene sets denoted in Figure 10D.

### 3.14. Essentiality of Genes

Out of a total of 7625 pan genes, 2064 (27%) genes maintain homology with 1415 unique DEG genes. About 924 out of 1906 (48%) persistent genes, 97 out of 626 (15%) shell genes, 286 out of 2134 (13%) cloud genes, and 757 out of 2945 (25%) unique genes exhibited homology with the DEG genes. A Venn diagram was constructed to visualize if there are genes in different sections of the pan-genome that show homology with the same DEG gene. It was observed that three DEG genes got hit by all sections of the pan-genome. The Venn diagram is demonstrated in Figure 10E. All genes of *A. muciniphila* ATCC BAA 835 were also inferred for homology with the DEG genes, and it was observed that out of 2243 genes, 955 were found to be homologous.

### 3.15. Genome Reduction Strategy

An important consideration in the genome reduction was to reduce redundancy and remove undesired genes. The shell, cloud, and unique genes are not present in all strains, so they are not necessary for the species. However, the persistent genes were evolutionarily maintained by almost all strains of the species despite multiple evolutionary pressures. Persistent genes are involved in the basic biology and fundamental processes of the species including metabolism and central dogma. These are considered crucial for metabolic potential and the long-term survival of a strain and species. In addition to persistent genes, every strain encodes additional genes that are unique to that strain. In *A. muciniphila*, 1906 are persistent genes, and on average 430 genes are non-persistent. Non-persistent genes may have arrived in the genomes by HGT or may be the result of gene loss from some of the strains of *A. muciniphila*. They may encode important pathways that provide advantage to some strains. In order to find a reduced genome that is stable and functional in the long run, a systematic approach was applied. The persistent genome was maintained in the reduced genome because the genes in the persistent genome have maintained themselves in the species despite many evolutionary pressures over a long period of time and possibly code for proteins involved in the fundamental processes such as the basic biology and phenotype of the species. Furthermore, the KEGG and EggNOG mapper indicated that persistent genes were enriched for functionally important metabolic pathways and involved in mechanisms vital to life such as replication, transcription, cell wall/membrane biogenesis, and translation. Moreover, the gene set was undergoing the lowest Ka/Ks ratio and maintained a stable GC content and average gene length. For the non-persistent gene set, the criterion for inclusion into the reduced genome was determined by essentiality analysis. If a gene was found to be essential after BLAST with the DEG, it was included. *A. muciniphila* ATCC BAA 835 was used as a chassis for the reduction criterion. Regions encoding functional genes, secondary metabolite gene clusters, KEGG orthologs, COG orthologs, and GO orthologs were not excluded. A landscape of necessary and unnecessary parts of the genome has been constructed after an extensive analysis of persistent genes, gene essentiality, and metabolic potential. The Circos plot depicts a visual representation of the above idea in Figure 11.

## 4. Discussion

Trillions of microbes are resident in the gastrointestinal tract. Many microbes are getting uprising attention as probiotics for their role in modulating disease and health in humans. *A. muciniphila* is a probiotic that is unique in its food source and found in both humans and animals. Its association with several metabolic and immunity-related diseases has been observed.

Previously, pan-genomics has been performed on a very limited number of genomes that provided a preliminary picture of the pan-genome dynamics of *A. muciniphila*. The recent boost in next-generation sequencing has resulted in the accumulation of sequences of various strains of *A. muciniphila* that require a more comprehensive analysis to understand the species diversity, metabolic potential, and evolutionary dynamics of the species [68]. This study comprehensively analyzed the population structure, genomic diversity, metabolic potential, and evolutionary parameters. Pan-genome was refined to entail only *A. miciniphila* genomes by systematically removing the sequences that had less ANI and contain an abnormally high number of unique genes. The study, further, integrated the persistent and essential genomes to come up with an idea of a minimal functional genome.

As a first step, ANI was used to exclude the genomes that are too diverse to be contained in the same species. This is the most important step to maintaining a stable and true representative landscape of the species pan-genome. Following ANI, 19 sequences were excluded from the downstream analysis, leaving 130 genomes for further analysis. Studies have suggested that the genomes below the 95% threshold influence the core genome disproportionately [69].

Previously, based on 39 genomes [70], the γ value was found to be 0.25; however, our study, based on 130 genomes, demonstrated that the genomes are expanding at double the rate that was previously speculated with a γ value of 0.49. Furthermore, each phylogroup was found to be independently expanding. Furthermore, the strains in each phylogroup are expanding independent of the within-group ANI range.

There was a disagreement in the number of phylogroups of *A. muciniphila* in different studies. Two studies based on 18 and 39 genomes have suggested three phylogroups [70,71] and one study based on 75 genomes suggested four phylogroups [72]. Recently, isolate genomes were combined with metagenomics to reveal the ecology of gut microbiota and a phylogenetic tree was constructed using core genes. The tree clustered *A. muciniphila* genomes into four phylogroups [73]. In the current study, different machine learning-based clustering approaches were employed to uncover the best potential phylogroups. PopPUNK was also used to uncover underlying clusters based on core genome distances and accessory genome distances. Furthermore, it was observed that clustering of gene presence-absence juxtaposed similarly with the PopPUNK clustering and machine learning approaches.

*A. muciniphila* is characterized by extensive recombination and a high rate of HGT. Both of these probably lead to an open pan-genome of the species. Furthermore, the highest recombination in AmIV possibly and at least partially answers the highest γ value in AmIV. HGT has been observed to be highest in the unique genome, followed by the cloud, shell, and persistent genomes. The HGT genes were shared to or from other gut microbes such as *Clostridiales, Bacteroidales, Firmicutes,* and *Akkermansiaceae.* As a probiotic, it was probable to have obtained genes from neighboring species. As expected, new genes incorporate regularly into the pan-genome as a result of HGT and lead to more diversity in the species. The persistent genome (core and soft-core genome) has the least fraction of HGT genes, which indicates that it contains species-specific gene content that is essential for the long-term benefit of the species.

*A. muciniphila* maintains genes involved in various important pathways such as SCFAs, various vitamins, antimicrobial metabolites, and other secondary metabolites. The presence of prophages, secondary metabolite gene clusters, and genomic islands have been observed in the representative strain of *A. muciniphila*.

The presence of antibiotic genes and pathogenic determinants raises concern for use as a probiotic. A special focus on the resistome profiling of probiotics has emerged recently. Studies have suggested the presence of genes conferring resistance to tetracycline, vancomycin, clindamycin, and ciprofloxacin in *Lactobacillus* and *Lactococcus,* [68,74] and dalfopristin in *E. faecalis* [75]. The HGT of antibiotic resistance genes to pathogens is also an emerging cause of concern for public health. The observation that 43% of the pan-genome is the result of HGT reflects that a high prevalence of HGT is prevailing in *A. muciniphila* with phylogenetically distant bacterial species. The presence of virulence factor is even more alarming and has very recently been profiled in various strains of *Lactobacillus* and *Bifidobacterium* [74,75,76].

The persistent genome was expectedly the most important in terms of metabolism. It maintained high GC content and average gene length as compared to other parts of the pan-genome, indicating the stability of the persistent genome. Meanwhile, Ka/Ks values of the persistent-genome are less than other sections of the pan-genome. It corresponds to the Ka/Ks values found previously [77]. Similarly, genes belonging to the persistent group underwent the least HGT.

Two important aspects of the study were to determine the evolutionary dynamics and design a reduced genome for next-generation probiotics. Pan-genome was systematically refined in the quest to get a more accurate prediction of both aspects. The study concluded a high rate of recombination within the different strains and HGT events from bacteria that colonize the gut. Lastly, based on the persistent genes, essential genes, and functionally annotated genes, a reduced genome was proposed for safer use as a next-generation probiotic. Recently, synthetic biology approaches have focused on the development of minimal or reduced genomes for many species such as *E. coli, Bacillus subtilis,* and *Corynebacterium glutamicum*. The reduced genomes provide the advantage of high production of desired compounds as compared to natural ones. In addition to the high yield of desired compounds, the small genomes provide the advantage of easier metabolic modeling and systematic prediction. However, minimal genomes are less robust to fluctuating conditions and may be unstable in the long run. Different methods have been proposed to tackle the problem of stability [78]. In the current analysis, we got the advantage of evolutionarily important genes that have survived in the whole species despite a variety of evolutionary pressures. These are the genes that are present in more than 95% of the genome [33]. Our analysis has confirmed that these genes are involved in important metabolic pathways that are fundamental to the functional properties of the species. We have built our genome reduction strategy on these persistent genes in an attempt to predict a reduced genome that is stable and represents the functional properties of the species. The approach will be useful for prediction of reduced genomes for related microbes.

## Figures and Tables

**Figure 1 microorganisms-10-01350-f001:**
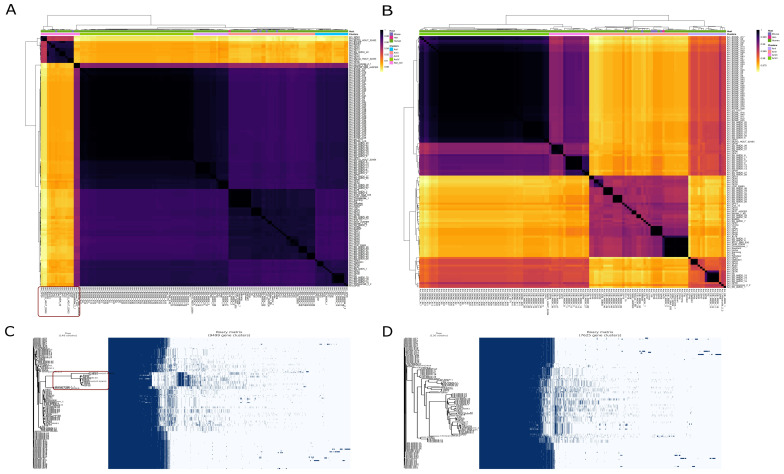
Average nucleotide identity and gene presence-absence profile of initial dataset (149 strains) and final dataset (130 strains). The strains that are too diverse to be incorporated in the final dataset are encircled in both ANI heat map and gene presence-absence profile. (**A**) Average nucleotide identity of 149 strains. A dark red rectangle encircles excluding 19 strains. (**B**) Average nucleotide identity of the final dataset (130) strains. (**C**) Gene presence-absence profile of 149 strains. The dark red rectangle encircles the gene presence-absence profile of the strains to be excluded. The high proportion of genes in these strains is different from the rest of the strains. (**D**) The gene presence-absence profile of 130 strains that were used for downstream analysis after exclusion of confounding strains.

**Figure 2 microorganisms-10-01350-f002:**
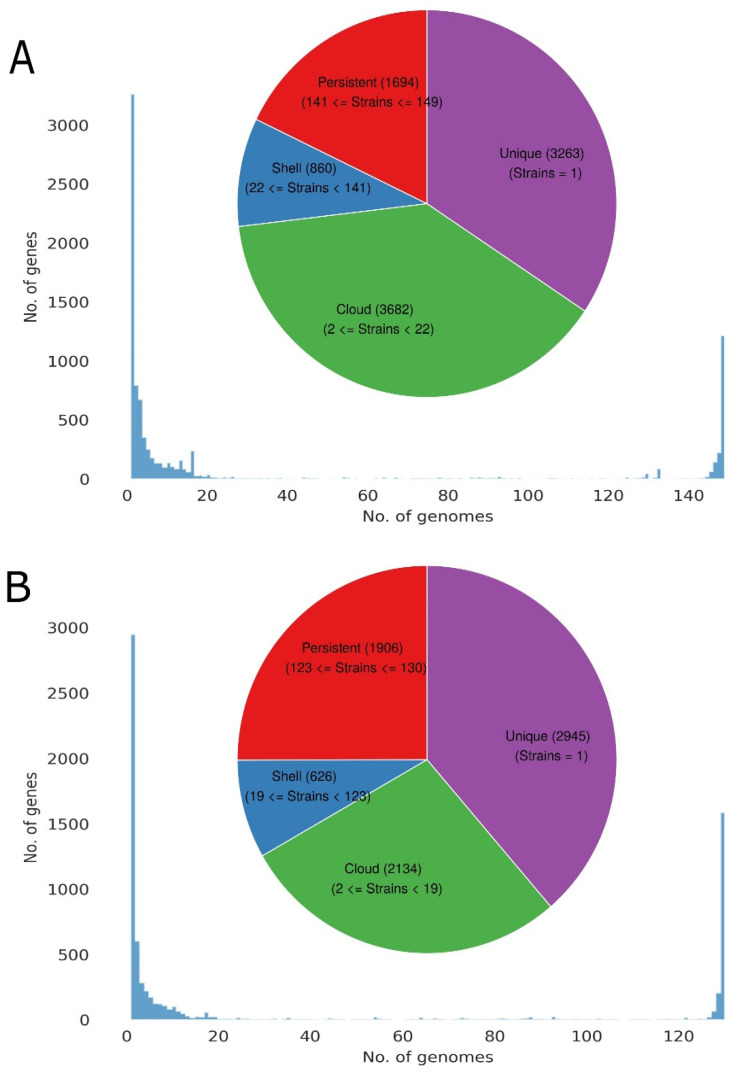
Distribution of pan-genome to the core, shell, cloud, and unique genome. (**A**) The pan-genome distribution of 149 strains. (**B**) The pan-genome distribution of 130 strains.

**Figure 3 microorganisms-10-01350-f003:**
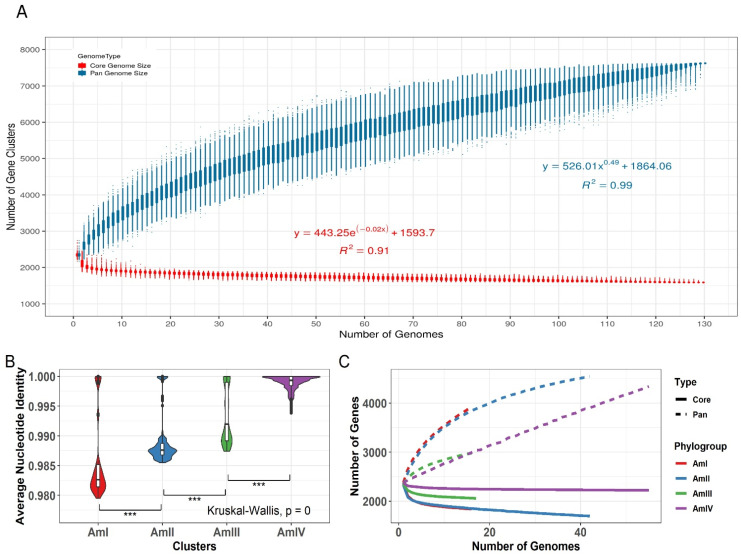
Pan-genome modeling of *A. muciniphila*. (**A**) Modelling of the pan-genome size of the 130 strains. It is evident that the pan-genome curve is rising upwards, indicating an expanding pan-genome. The γ of more than 0 indicates an open genome, and the correct data revealed a γ value of 0.49. (**B**) The range of ANI across four phylogroups. AmI shows the most variability in ANI among all the phylogroups, while strains of the AmIV are more similar to each other as compared to that of other phylogroups. (**C**) The pan-genome modeling of all the subgroups of *A. muciniphila.* The γ values were 0.31, 0.17, 0.2, and 0.9 for AmI, AmII, AmIII, and AmIV, respectively. The highest number of genes are predicted to be in AmII, while AmIV is expanding at a rate higher than others including AmII. Significance is denoted by: “***” if *p* value is less than 0.001.

**Figure 4 microorganisms-10-01350-f004:**
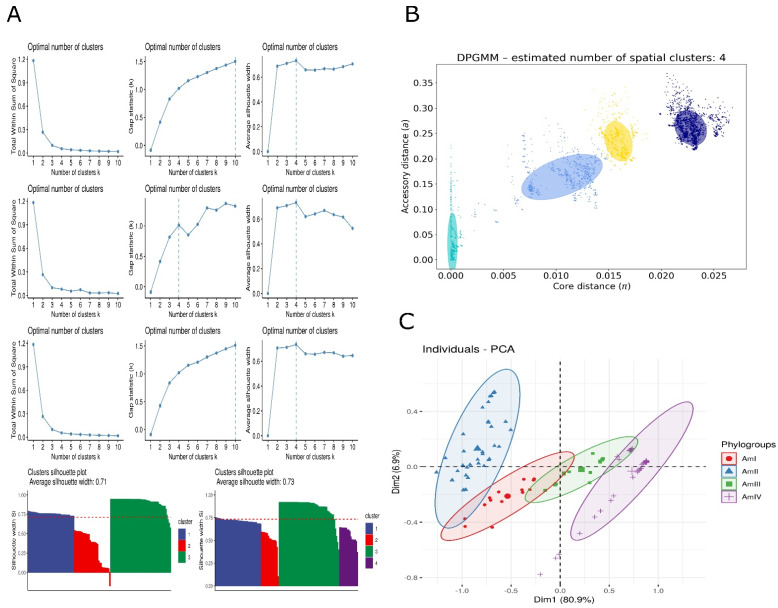
Clustering of strains into phylogroups. (**A**) (I–III) Partition around medoids clustering prediction using (I) within sum of square, (II) gap distances, and (III) silhouette distances. (IV–VI) Kmeans clustering prediction using (IV) within sum of square, (V) gap distances, and (VI) silhouette distances. (VII–IX) Hierarchical clustering prediction using (VII) within sum of square, (VIII) gap distances (IX), and silhouette distances. (X) Silhouette plot for three clusters. (XI) Silhouette plot for four clusters. (**B**) Dirichlet Process Gaussian Mixture Model-based clustering of strains into phylogroups using core distances and accessory distances. (**C**) Principal component analysis of gene presence-absence profile.

**Figure 5 microorganisms-10-01350-f005:**
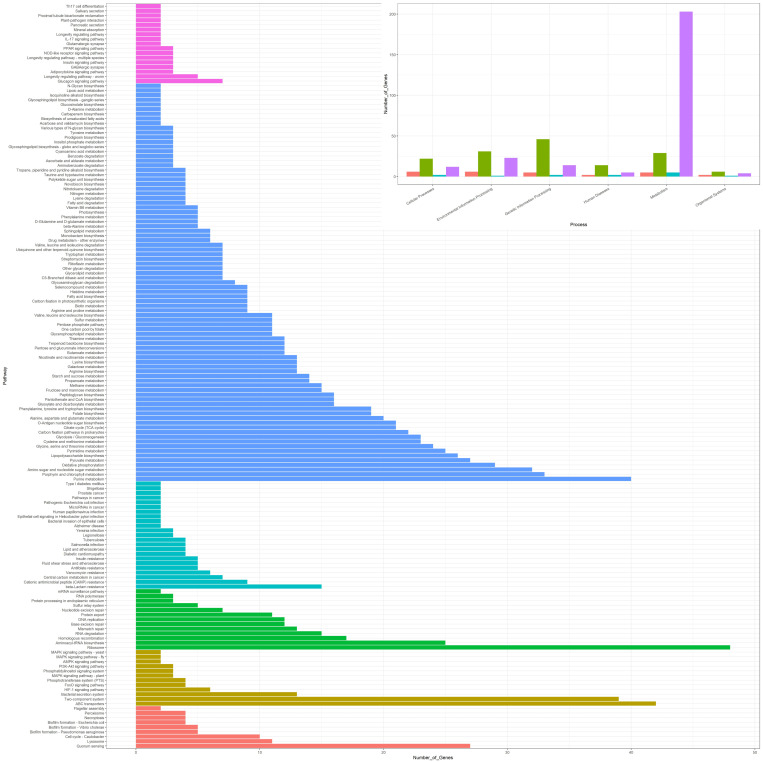
KEGG pathway annotation by KAAS. The majority of genes in the genetic information processing, environmental information processing, and cellular processes belong to the core genome. Furthermore, the species has been found to carry diverse metabolic potential and is involved in many important biological pathways such as vitamin synthesis and synthesis of secondary metabolites.

**Figure 6 microorganisms-10-01350-f006:**
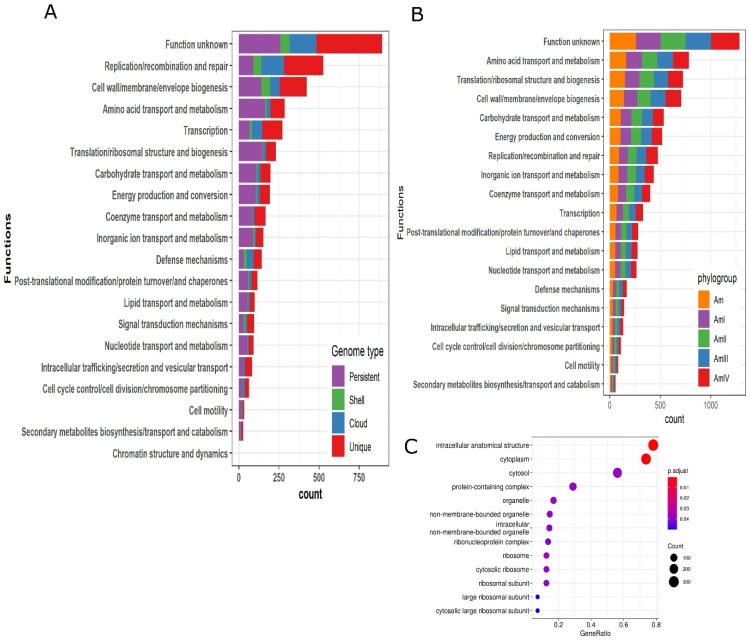
Clusters of orthologous (COG) annotation, enrichment analysis, and genomic islands. (**A**) The functions are arranged according to the number of genes related to that function. The top five functional categories are “Function unknown”, “Replication/recombination and repair”, “Cell wall/membrane/envelop biogenesis”, “Amino acid transport and metabolism”, and “Transcription”. (**B**) Comparison of the core genome of all strains (130) and core genome of all the phylogroups. (**C**) Enrichment analysis of all subsections of the pan-genome was performed. Only persistent genome expressed enrichment in different pathways. The enrichment was related to pathways such as intracellular anatomical structure, cytoplasm, cytosol, and protein-containing complex, etc.

**Figure 7 microorganisms-10-01350-f007:**
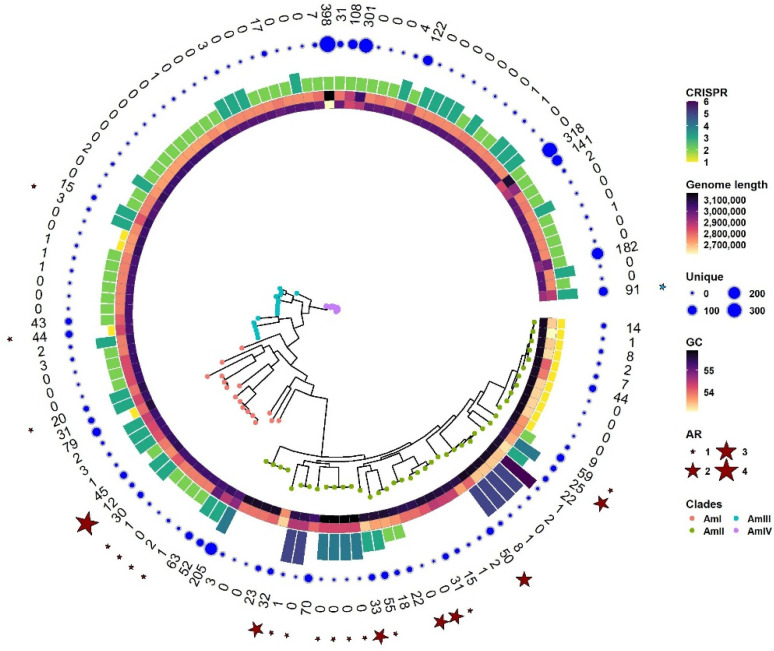
Phylogenetic tree of *A. muciniphila* strains. Information around the tree provides an overview of the genetic potential of different strains predicted using suitable tools. Strains from different phylogroups are colored differently. The clustering of the strains in the phylogenetic tree corresponds to the clustering inferred in Section 3.4. Layers around the tree denote (from inside out) GC content, genome length, number of CRISPR sequences, number of unique genes as a bubble, number of unique genes in digits, antibiotic resistance genes, and virulent factors. Antibiotic resistance genes are shown by red stars while virulent factors are shown by green stars.

**Figure 8 microorganisms-10-01350-f008:**
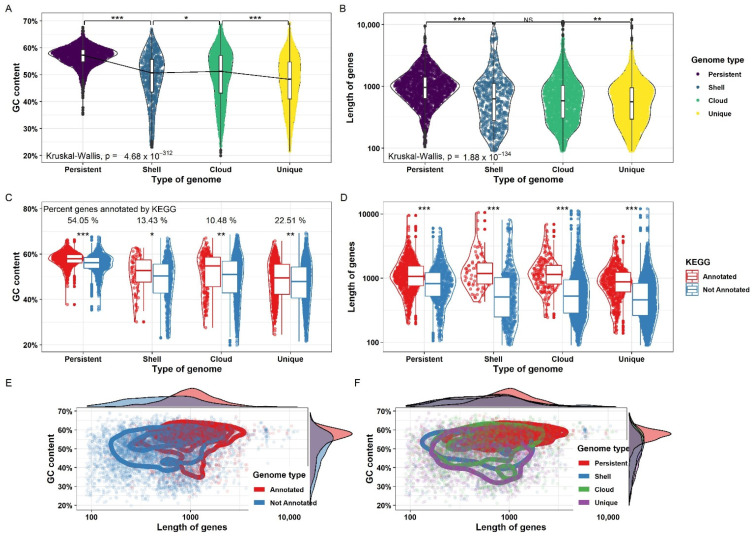
Distribution of GC content and gene length among different sections of the pan-genome. (**A**) Distribution of GC content in the persistent, cloud, shell, and unique genome. The difference in the GC content among different sections was found to be significant with a very low *p*-value. (**B**) Distribution of gene length across different sub-sections of the pan-genome. The difference is non-significant between the cloud and shell genomes while significant for others. (**C**) GC content across different sub-sections of the pan-genome is further divided into genes that are annotated by KAAS and those that are not annotated by KAAS within each sub-section. The difference in GC content between annotated and non-annotated genes was found to be significant within each group. (**D**) Gene length across different sub-sections of the pan-genome is further divided into genes that are annotated by KAAS and those that are not annotated by KAAS within each sub-section. The difference in gene length between annotated and non-annotated genes was found to be significant within each group. (**E**,**F**) GC content and gene length was plotted for annotated and non-annotated genes and across all sub-sections of the pan-genome. It was observed that annotated genes tend to have higher GC content and gene length. Similarly, GC content and gene length tended to be for the persistent genome as compared to the cloud, shell, and unique genomes. Significance is denoted by: “***” if *p* value is less than 0.001, “**” if *p* value is less than 0.01 and more than 0.001 and “*” if *p* value is less than 0.05 and more than 0.01.

**Figure 9 microorganisms-10-01350-f009:**
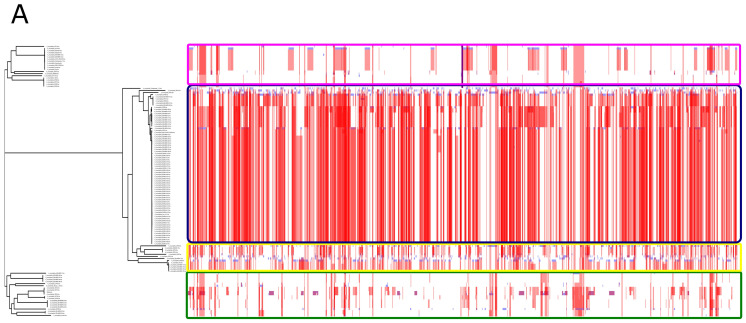
Evolutionary dynamics of *A. muciniphila* depicting recombination. (**A**) Recombination among different strains as depicted by Gubbins. Based on the recombination pattern, it is evident that all phylogroups share recombination as indicated by the red tiles. (**B**) Network graph of the recombination pattern showed two distinct clusters. One cluster contains strains from all the phylogroups while one cluster consists of some strains of AmII and one strain of AmIV.

**Figure 10 microorganisms-10-01350-f010:**
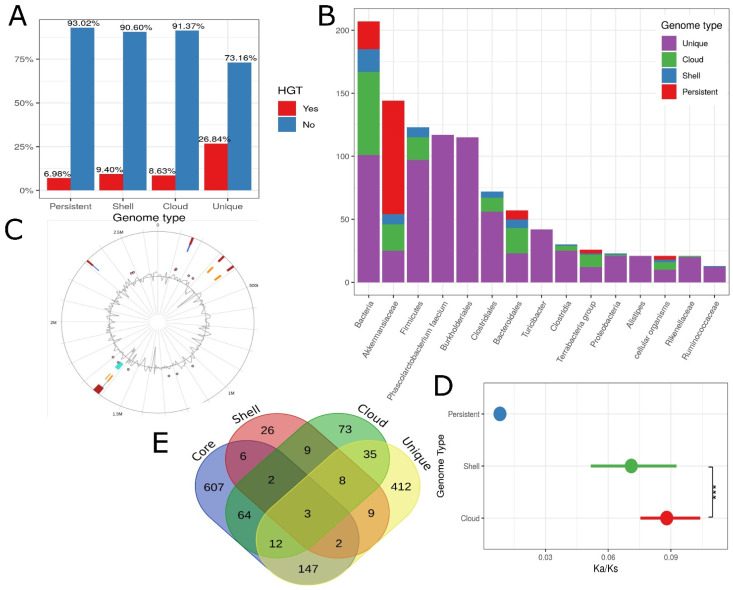
(**A**) Horizontal gene transfer prediction by HGTector within each sub-section of pan-genome. (**B**) The taxonomic groups that are predicted to share horizontally transferred genes with *A. muciniphila*. It has been observed that the majority of taxonomic groups reside in the gastrointestinal tract. (**C**) Genomic Islands as indicated by IslandViewer4. (**D**) Ka/Ks in the genes belonging to the persistent, cloud, and shell genomes. Genes that belong to the persistent genome showed the least values of Ka/Ks followed by the cloud and shell genomes. (**E**) Essentiality analysis of the pan-genome. The overlap shows the common essential gene that reveals high similarity with genes from more than one sub-section of the pan-genome. Significance is denoted by: “***” if *p* value is less than 0.001.

**Figure 11 microorganisms-10-01350-f011:**
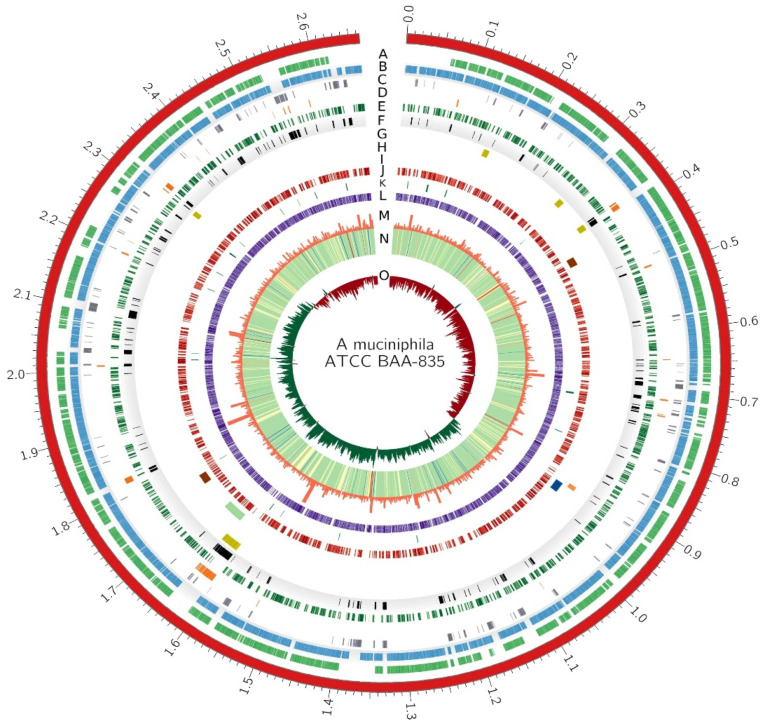
Circos plot of *A. muciniphila ATCC BAA 835*. Different layers of the circos are represented by A–O. A—genes that are present in 100% of the genomes, B—genes that are present in 95% of the genomes, C—shell genome, D—unique genome, E—essential genes, F—undesired genes (those that are neither essential nor persistent, G—genomic islands, H—phages as depicted by PHASTER, I—secondary metabolite gene clusters by AntiSMASH, J—genes annotated by KEGG, K—genes annotated by GO, L—genes annotated by COG, M—histogram of gene length, N—heatmap of GC content, O—histogram of GC-skew.

## Data Availability

The dataset analyzed for this study can be found in the NCBI Genome repository [https://www.ncbi.nlm.nih.gov/genome/browse/#!/prokaryotes/1598/] accessed on 24 March 2021.

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
