# Peer review of "Comparative Genomics and Pan-Genome Driven Prediction of a Reduced Genome of Akkermansia muciniphila"

_microorganisms, 2022, doi:10.3390/microorganisms10071350_

Round 1
Reviewer 1 Report
Bukhari et al. provide a detailed analysis of the pangenome of Akkermansia muciniphila. Their work nearly duplicates the number of genomes studied and define a minimal genome for this species. The work is relevant but the quality of the figures hinders the evaluation of the data presented. I strongly recommend to improve the quality of the images.
1. Labels, graph axis and text in figures is hard to read in their current version. A higher resolution version of the figures in the manuscript would greatly help solve this issue or resizing the text to a larger size. Figure 1 could use some polishing, as details from the graph that cannot be read like the legends can be removed. Text in Figure 5 cannot be read at all, so the categories in it cannot be distinguished. Not all panels in the figures are mentioned in the text.
2. Figure 3 comes after figure 4.
3. Description of panel B is missing in Figure 4.
4. Supplementary material was not available for review
Author Response
Bukhari et al. provide a detailed analysis of the pangenome of Akkermansia muciniphila. Their work nearly duplicates the number of genomes studied and define a minimal genome for this species. The work is relevant but the quality of the figures hinders the evaluation of the data presented. I strongly recommend to improve the quality of the images.
Response: Grateful for your suggestions, however, the analysis is unique in prediction of reduced genome of A. muciniphila. Population structure has been elucidated using variety of different methods in order to deal with ambiguities.
- Labels, graph axis and text in figures is hard to read in their current version. A higher resolution version of the figures in the manuscript would greatly help solve this issue or resizing the text to a larger size. Figure 1 could use some polishing, as details from the graph that cannot be read like the legends can be removed. Text in Figure 5 cannot be read at all, so the categories in it cannot be distinguished. Not all panels in the figures are mentioned in the text.
Response: Higher resolution figures are being provided separately in JPEG. Due to the size of Figure 1 and Figure 5, both are also provided in .svg format as suggested by other reviewers as well.
- Figure 3 comes after figure 4.
Response: The Figures were mislabeled, the corrections have been made.
- Description of panel B is missing in Figure 4.
Response:Description of Figure 4 panel B has been added.
- Supplementary material was not available for review
Response: Supplementary material has been added.
Reviewer 2 Report
- First of all, I would like to felicitate the authors for such immense work and how the analyses were addressed. I believed that the aim of this study was accomplished and novel.
- Due to the large size of information in figure 1, I would recommend being more specific within the description and results of the ANI comparison (lines 208-219). Instead of saying Figure 1 in line 219, separate in A, B, C, and D to be clearer with the figure description. Also, I would recommend using a .SVG format for these types of figures.
-line 253: Figure 4 needs to be changed for Figure 3 that it is 273 (it seems that the figures are mislabelled, please review)
-Figure 5 is not readable; please change to .SVG format
-In point 3.8, you jumped from Figure 6 to Figure 10 (C); what about the description of Figure 7 and Figure 8. It might be necessary to change the order of the description of the results. The description needs to follow the order of the figures to be easy to read and follow.
-I believe that figure 7 shows a great amount of work and data collected, however, I think that the pink bubbles behind make a huge background noise to the figure, in contrast, a white background would be more readable and looks better.
Author Response
Comments and Suggestions for Authors
- First of all, I would like to felicitate the authors for such immense work and how the analyses were addressed. I believed that the aim of this study was accomplished and novel.
Thank you so much for your encouraging comments.
- Due to the large size of information in figure 1, I would recommend being more specific within the description and results of the ANI comparison (lines 208-219). Instead of saying Figure 1 in line 219, separate in A, B, C, and D to be clearer with the figure description. Also, I would recommend using a .SVG format for these types of figures.
Response: The description has been modified for clear understanding. However, both ANI and gene presence absence were used for screening out divergent genomes. A and C are before removal of 19 genomes and B and D are after that. One panel was not used for removal of genomes so we may not describe one panel on line 219. Figure legend further explains the strategy. Yes, the figure is big so an .SVG has also been provided.
-line 253: Figure 4 needs to be changed for Figure 3 that it is 273 (it seems that the figures are mislabelled, please review)
Response: The correction has been added. It was mislabeled.
-Figure 5 is not readable; please change to .SVG format
Response: A higher resolution jpeg and svg has been added.
-In point 3.8, you jumped from Figure 6 to Figure 10 (C); what about the description of Figure 7 and Figure 8. It might be necessary to change the order of the description of the results. The description needs to follow the order of the figures to be easy to read and follow.
Response: The suggested rearrangements have been made.
-I believe that figure 7 shows a great amount of work and data collected, however, I think that the pink bubbles behind make a huge background noise to the figure, in contrast, a white background would be more readable and looks better.
Response: Bubble has been removed.
Reviewer 3 Report
Title:
"Comparative genomics and Pan-genome driven prediction of a 2 reduced genome of Akkermansia muciniphila"
The manuscript describes the pan-genome analysis of the Akkermanisia muciniphila in 130 genomes. Moreover, the authors attempted to elucidate evolutionary patterns underlying diversification of A. muciniphilaspecies by using computer-based analysis methods.
The result is interesting, but a considerable revision of manuscript is needed before it can be accepted for publication. The main problem with the paper is the description of the results and discussion of their significance.
Before final decision by editor, there are several points that should be addressed by the author.
Suggested changes and questions are listed below:
1. Page 2, lines 70-116:
The writing is very diffuse and lacks conciseness. This section needs to be re-written to increase clarity and sharpness.
2. Page 4, line 153:
"Fasttree" should read "FastTree 2".
3. Page 4, lines 186-188:
The authors should provide references for the Gubbins and HGTector.
4. Page 5, lines 212-219:
It's not clear what is meant by these sentences. This needs to be clarified.
5. Page 7, line 254
"Figure 4" should read "Figure 3"
6. Page 7, lines 254-259:
There is no description (Legend) for B).
7. Page 8, line 274:
"Figure 3" should read "Figure 4"
8. Page 8, line 280:
"Principles component" should read "Principal component".
8. Page 8, lines 300-324:
The writing is diffusive and obscure. Moreover, pathway terms in the Figure 5
are so small that it is barely readable. This section needs to be re-written and Figure 5 should be remake.
9. Page 11, lines 361-378:
The description of this section is difficult to follow, because original data or Figure is not clearly related. In addition, I do not understand the meaning of Figure 7. These need to be clarified.
10. Page 13, lines 424-435:
I feel the writing is incomplete and diffusive. Moreover, it's not clear what is meant by Figure 9. It needs to describe Figure 9 in detail and this section needs to be re-written.
11. Page 15, lines 460-466:
In order to demonstrate positive selection, it is essential to indicate the significance of positive selection statistically. However, the authors only calculated Ka/Ks values and used as a measure of positive selection. If the authors need to show and discuss positive selection, it needs to conduct a more elaborated method such as Likelihood ratio tests for positive selection of individual genes.
12. Page 15, lines 467-509:
The writing is very diffusive and the meaning of the analysis in this section is hard to understand. Moreover, Figure 11 is not clearly related to the description. This section needs to be re-written and the meaning of the analysis and its results should be clearly described.
13. Page 17, lines 510-584:
I found the authors' explanations in the Discussion are also very diffusive and obscure.
The wording and style in the Discussion, need careful editing and clarification.
14.It may be appropriate to include recent references regarding the comparative analysis of Akkermansia muciniphila genomes indicated below.
(1) Genomic diversity and ecology of human-associated Akkermansia species in the gut microbiome revealed by extensive metagenomic assembly.
N Karcher, E Nigro, M Punčochář, A Blanco-Míguez, Genome Biology, 2021
(2)The evolution and competitive strategies of Akkermansia muciniphila in gut.
JS Kim, SW Kang, JH Lee, SH Park, JS Lee - Gut microbes, 2022 - Taylor & Francis
15. There are many ambiguous wordings and use of grammar. I do suggest to have some external language editing done by a person familiar with the field.
Author Response
Suggested changes and questions are listed below:
- Page 2, lines 70-116:
The writing is very diffuse and lacks conciseness. This section needs to be re-written to increase clarity and sharpness.
Response: The section has been modified
- Page 4, line 153:
"Fasttree" should read "FastTree 2".
Response: The suggested change has been introduced
- Page 4, lines 186-188:
The authors should provide references for the Gubbins and HGTector.
Response: Reference has been introduced.
- Page 5, lines 212-219:
It's not clear what is meant by these sentences. This needs to be clarified.
Response: The sentences have been modified for clear understanding
- Page 7, line 254
"Figure 4" should read "Figure 3"
Response: Yes it was mislabeled, change has been made.
- Page 7, lines 254-259:
There is no description (Legend) for B).
Response: The legend for the Panel B has been introduced.
- Page 8, line 274:
"Figure 3" should read "Figure 4"
Response: Yes it was mislabeled, change has been made.
- Page 8, line 280:
"Principles component" should read "Principal component".
Response: Changes has been made.
- Page 8, lines 300-324:
The writing is diffusive and obscure. Moreover, pathway terms in the Figure 5 are so small that it is barely readable. This section needs to be re-written and Figure 5 should be remake.
Response: A higher resolution figure in jpg and svg has been provided.
- Page 11, lines 361-378:
The description of this section is difficult to follow, because original data or Figure is not clearly related. In addition, I do not understand the meaning of Figure 7. These need to be clarified.
Response: The section has been split into two different sections. The purposes of the figure 7 are i) a phylogenetic tree of all 130 strains colored by phylogroups, ii) description of genomic features of each strain such as GC content, CRISPR sequences, antibiotic resistance genes. We found that the prevalence of antibiotic genes is less in AmIV and AmIII. Bubble has been removed that might have been causing background noise.
- Page 13, lines 424-435:
I feel the writing is incomplete and diffusive. Moreover, it's not clear what is meant by Figure 9. It needs to describe Figure 9 in detail and this section needs to be re-written.
Response: The figure has been described. And section has been modified for clear understanding.
- Page 15, lines 460-466:
In order to demonstrate positive selection, it is essential to indicate the significance of positive selection statistically. However, the authors only calculated Ka/Ks values and used as a measure of positive selection. If the authors need to show and discuss positive selection, it needs to conduct a more elaborated method such as Likelihood ratio tests for positive selection of individual genes.
Response: For positive selection, Ka/Ks calculator was used that performs Fishers test for statistical significance. However, p values for many genes was found to be NA. So the section has been modified to describe Ka/Ks ratio rather than positive selection. It would not impact the main conclusion.
- Page 15, lines 467-509:
The writing is very diffusive and the meaning of the analysis in this section is hard to understand. Moreover, Figure 11 is not clearly related to the description. This section needs to be re-written and the meaning of the analysis and its results should be clearly described.
Response: Figure 11 is the summary of a major portion of the analysis. Furthermore, it describes the undesired genes in an effort towards a reduced genome. All the layers of the figure has been labelled as A to O and described appropriately in figure legend.
- Page 17, lines 510-584:
I found the authors' explanations in the Discussion are also very diffusive and obscure.
The wording and style in the Discussion, need careful editing and clarification.
Response: Efforts has been made to make writing as clear as possible for a reader.
14.It may be appropriate to include recent references regarding the comparative analysis of Akkermansia muciniphila genomes indicated below.
(1) Genomic diversity and ecology of human-associated Akkermansia species in the gut microbiome revealed by extensive metagenomic assembly.
N Karcher, E Nigro, M Punčochář, A Blanco-Míguez, Genome Biology, 2021
(2)The evolution and competitive strategies of Akkermansia muciniphila in gut.
JS Kim, SW Kang, JH Lee, SH Park, JS Lee - Gut microbes, 2022 - Taylor & Francis
Response: References has been introduced appropriately.
- There are many ambiguous wordings and use of grammar. I do suggest to have some external language editing done by a person familiar with the field.
Response: All the manuscript has been reread and changes has been made.
Round 2
Reviewer 3 Report
There are still a number of odd wordings and typographical errors throughout the manuscript. English of the manuscript should be checked extensively.
Author Response
The manuscript is carefully revised as per instructions